# DON'T SETTLE FOR AVERAGE, GO FOR THE MAX: FUZZY SETS AND MAX-POOLED WORD VECTORS

**Vitalii Zhelezniak, Aleksandar Savkov, April Shen, Francesco Moramarco,**
**Jack Flann & Nils Y. Hammerla**
Babylon Health
{vitali.zhelezniak, sasho.savkov, firstname.lastname}
@babylonhealth.com

## ABSTRACT

Recent literature suggests that averaged word vectors followed by simple post-processing outperform many deep learning methods on semantic textual similarity tasks. Furthermore, when averaged word vectors are trained supervised on large corpora of paraphrases, they achieve state-of-the-art results on standard STS benchmarks. Inspired by these insights, we push the limits of word embeddings even further. We propose a novel fuzzy bag-of-words (FBoW) representation for text that contains all the words in the vocabulary simultaneously but with different degrees of membership, which are derived from similarities between word vectors. We show that max-pooled word vectors are only a special case of fuzzy BoW and should be compared via fuzzy Jaccard index rather than cosine similarity. Finally, we propose DynaMax, a completely unsupervised and non-parametric similarity measure that dynamically extracts and max-pools good features depending on the sentence pair. This method is both efficient and easy to implement, yet outperforms current baselines on STS tasks by a large margin and is even competitive with supervised word vectors trained to directly optimise cosine similarity.

## 1 INTRODUCTION

Natural languages are able to encode sentences with similar meanings using very different vocabulary and grammatical constructs, which makes determining the semantic similarity between pieces of text a challenge. It is common to cast semantic similarity between sentences as the proximity of their vector representations. More than half a century since it was first proposed, the Bag-of-Words (BoW) representation (Harris, 1954; Salton et al., 1975; Manning et al., 2008) remains a popular baseline across machine learning (ML), natural language processing (NLP), and information retrieval (IR) communities. In recent years, however, BoW was largely eclipsed by representations learned through neural networks, ranging from shallow (Le & Mikolov, 2014; Hill et al., 2016) to recurrent (Kiros et al., 2015; Conneau et al., 2017; Subramanian et al., 2018a), recursive (Socher et al., 2013; Tai et al., 2015), convolutional (Kalchbrenner et al., 2014; Kim, 2014), self-attentive (Vaswani et al., 2017; Cer et al., 2018a) and hybrid architectures (Gan et al., 2017; Tang et al., 2017; Zhelezniak et al., 2018).

Interestingly, Arora et al. (2017) showed that averaged word vectors (Mikolov et al., 2013a; Pennington et al., 2014; Bojanowski et al., 2016; Joulin et al., 2017) weighted with the Smooth Inverse Frequency (SIF) scheme and followed by a Principal Component Analysis (PCA) post-processing procedure were a formidable baseline for Semantic Textual Similarity (STS) tasks, outperforming deep representations. Furthermore, Wieting et al. (2015; 2016) and Wieting & Gimpel (2018) showed that averaged word vectors trained supervised on large corpora of paraphrases achieve state-of-the-art results, outperforming even the supervised systems trained directly on STS.

Inspired by these insights, we push the boundaries of word vectors even further. We propose a novel fuzzy bag-of-words (FBoW) representation for text. Unlike classical BoW, fuzzy BoW contains all the words in the vocabulary simultaneously but with different degrees of membership, which are derived from similarities between word vectors.

Next, we show that max-pooled word vectors are a special case of fuzzy BoW. Max-pooling significantly outperforms averaging on standard benchmarks when word vectors are trained unsupervised.

Since max-pooled vectors are just a special case of fuzzy BoW, we show that the fuzzy Jaccard index is a more suitable alternative to cosine similarity for comparing these representations. By contrast, the fuzzy Jaccard index completely fails for averaged word vectors as there is no connection between the two. The max-pooling operation is commonplace throughout NLP and has been successfully used to extract features in supervised systems (Collobert et al., 2011; Kim, 2014; Kenter & de Rijke, 2015; De Boom et al., 2016; Conneau et al., 2017; Dubois, 2017; Shen et al., 2018); however, to the best of our knowledge, the present work is the first to study max-pooling of pre-trained word embeddings in isolation and to suggest theoretical underpinnings behind this operation.

Finally, we propose DynaMax, a completely unsupervised and non-parametric similarity measure that dynamically extracts and max-pools good features depending on the sentence pair. DynaMax outperforms averaged word vector with cosine similarity on every benchmark STS task when word vectors are trained unsupervised. It even performs comparably to Wieting & Gimpel (2018)'s vectors under cosine similarity, which is a striking result as the latter are in fact trained supervised to directly optimise cosine similarity between paraphrases, while our approach is completely unrelated to that objective. We believe this makes DynaMax a strong baseline that future algorithms should aim to beat in order to justify more complicated approaches to semantic similarity.

As an additional contribution, we conduct significance analysis of our results. We found that recent literature on STS tends to apply unspecified or inappropriate parametric tests, or leave out significance analysis altogether in the majority of cases. By contrast, we rely on nonparametric approaches with much milder assumptions on the test statistic; specifically, we construct bias-corrected and accelerated (BCa) bootstrap confidence intervals (Efron, 1987) for the delta in performance between two systems. We are not aware of any prior works that apply such methodology to STS benchmarks and hope the community finds our analysis to be a good starting point for conducting thorough significance testing on these types of experiments.

## 2 SENTENCES AS FUZZY SETS

The bag-of-words (BoW) model of representing text remains a popular baseline across ML, NLP, and IR communities. BoW, in fact, is an extension of a simpler set-of-words (SoW) model. SoW treats sentences as sets, whereas BoW treats them as multisets (bags) and so additionally captures how many times a word occurs in a sentence. Just like with any set, we can immediately compare SoW or BoW using set similarity measures (SSMs), such as

$$\text{Jaccard}(A, B) = \frac{|A \cap B|}{|A \cup B|}, \quad \text{Otsuka}(A, B) = \frac{|A \cap B|}{\sqrt{|A| \times |B|}}, \text{ and } \quad \text{Dice}(A, B) = \frac{2|A \cap B|}{|A| + |B|}.$$

These coefficients usually follow the pattern $\frac{\#\{\text{shared elements}\}}{\#\{\text{total elements}\}}$. From this definition, it is clear that sets with no shared elements have a similarity of $0$, which is undesirable in NLP as sentences with completely different words can still share the same meaning. But can we do better?

For concreteness, let's say we want to compare two sentences corresponding to the sets $A = \{\text{'he', 'has', 'a', 'cat'}\}$ and $B = \{\text{'she', 'had', 'one', 'dog'}\}$. The situation here is that $A \cap B = \varnothing$ and so their similarity according to any SSM is $0$. Yet, both $A$ and $B$ describe pet ownership and should be at least somewhat similar. If a set contains the word 'cat', it should also contain a bit of 'pet', a bit of 'animal', also a little bit of 'tiger' but perhaps not too much of an 'airplane'. If both $A$ and $B$ contained 'pet', 'animal', etc. to *some degree*, they would have a non-zero similarity.

This intuition is the main idea behind fuzzy sets: a fuzzy set includes *all* words in the vocabulary simultaneously, just with different degrees of membership. This generalises classical sets where a word either belongs to a set or it doesn't.

We can easily convert a singleton set such as $\{\text{'cat'}\}$ into a fuzzy set using a similarity function $\text{sim}(w_i, w_j)$ between words. We simply compute the similarities between 'cat' and all the words $w_j$ in the vocabulary and treat those values as membership degrees. As an example, the set $\{\text{'cat'}\}$ really becomes $\{\text{'cat'} : 1, \text{'pet'} : 0.9, \text{'animal'} : 0.85, \ldots, \text{'airplane'} : 0.05, \ldots\}$

Fuzzifying singleton sets is straightforward, but how do we go about fuzzifying the entire sentence $\{\text{'he', 'has', 'a', 'cat'}\}$? Just as we use the classical union operation $\cup$ to build bigger sets from smaller ones, we use the *fuzzy union* to do the same but for fuzzy sets. The membership degree of a

word in the fuzzy union is determined as the *maximum* membership degree of that word among each of the fuzzy sets we want to unite. This might sound somewhat arbitrary: after all, why $\max$ and not, say, sum or average? We explain the rationale in Section 2.1; and in fact, we use the $\max$ for the classical union all the time without ever noticing it. Indeed, $\{\text{'cat'}\} \cup \{\text{'cat'}\} = \{\text{'cat'}\}$ and not $\{\text{'cat'} : 2\}$. This is simply because we computed $\max(1,1) = 1$ and not $\text{sum}(1,1) = 2$. Similarly $\{\text{'cat'}\} \cup \varnothing = \{\text{'cat'}\}$ since $\max(1,0) = 1$ and not $\text{avg}(1,0) = 1/2$.

The key insight here is the following. An object that assigns the degrees of membership to words in a fuzzy set is called the membership function. Each word defines a membership function, and even though 'cat' and 'dog' are different, they are semantically similar (in terms of cosine similarity between their word vectors, for example) and as such give rise to very similar membership functions. This functional proximity will propagate into the SSMs, thus rendering them a much more realistic model for capturing semantic similarity between sentences. To actually compute the fuzzy SSMs, we need just a few basic tools from fuzzy set theory, all of which we briefly cover in the next section.

## 2.1 FUZZY SETS: THE BARE MINIMUM

Fuzzy set theory (Zadeh, 1996) is a well-established formalism that extends classical set theory by incorporating the idea that elements can have degrees of membership in a set. Constrained by space, we define the bare minimum needed to compute the fuzzy set similarity measures and refer the reader to Klir et al. (1997) for a much richer introduction.

**Definition:** A set of all possible terms $\mathbb{V} = \{w_1, w_2, \ldots, w_N\}$ that occur in a certain domain is called a universe.

**Definition:** A function $\mu : \mathbb{V} \to \mathbb{L} \subseteq \mathbb{R}$ is called a membership function.

**Definition:** A pair $A = (\mathbb{V}, \mu)$ is called a fuzzy set.

Notice how the above definition covers all the set-like objects we discussed so far. If $\mathbb{L} = \{0, 1\}$, then $A$ is simply a classical set and $\mu$ is its indicator (characteristic) function. If $\mathbb{L} = \mathbb{N}^{\geq 0}$ (non-negative integers), then $A$ is a multiset (a bag) and $\mu$ is called a count (multiplicity) function. In literature, $A$ is called a fuzzy set when $\mathbb{L} = [0, 1]$. However, we make no restrictions on the range and call $A$ a fuzzy set even when $\mathbb{L} = \mathbb{R}$, i.e. all real numbers.

**Definition:** Let $A = (\mathbb{V}, \mu)$ and $B = (\mathbb{V}, \nu)$ be two fuzzy sets. The union of $A$ and $B$ is a fuzzy set $A \cup B = (\mathbb{V}, \max(\mu, \nu))$. The intersection of $A$ and $B$ is a fuzzy set $A \cap B = (\mathbb{V}, \min(\mu, \nu))$.

Interestingly, there are many other choices for the union and intersection operations in fuzzy set theory. However, only the $\max$-$\min$ pair makes these operations idempotent, i.e. such that $A \cup A = A$ and $A \cap A = A$, just as in the classical set theory. By contrast, it is not hard to verify that neither sum nor average satisfy the necessary axioms to qualify as a fuzzy union or intersection.

**Definition:** Let $A = (\mathbb{V}, \mu)$ be a fuzzy set. The number $|A| = \sum_{w \in \mathbb{V}} \mu(w)$ is called the cardinality of a fuzzy set.

Fuzzy set theory provides a powerful framework for reasoning about sets with uncertainty, but the specification of membership functions depends heavily on the domain. In practice these can be designed by experts or learned from data; below we describe a way of generating membership functions for text from word embeddings.

## 2.2 FUZZY BAG-OF-WORDS

From the algorithmic point of view any bag-of-words is just a row vector. The $i$-th term in the vocabulary has a corresponding $N$-dimensional one-hot encoding $\boldsymbol{e}^{(i)}$. The vectors $\boldsymbol{e}^{(i)}$ are orthonormal and in totality form the standard basis of $\mathbb{R}^N$. The BoW vector for a sentence $S$ is simply $\boldsymbol{b}^S = \sum_{i=1}^{N} c_i \boldsymbol{e}^{(i)}$, where $c_i$ is the count of the word $w_i$ in $S$.

The first step in creating the fuzzy BoW representation is to convert every term vector $\boldsymbol{e}^{(i)}$ into a membership vector $\boldsymbol{\mu}^{(i)}$. It really is the same as converting a singleton set $\{w_i\}$ into a fuzzy set. We call this operation 'word fuzzification', and in the matrix form it is simply written as

---

**Algorithm 1** DynaMax-Jaccard

---

**Input:** Word embeddings for the first sentence $\boldsymbol{x}^{(1)}, \boldsymbol{x}^{(2)} \ldots, \boldsymbol{x}^{(k)} \in \mathbb{R}^{1 \times d}$
**Input:** Word embeddings for the second sentence $\boldsymbol{y}^{(1)}, \boldsymbol{y}^{(2)} \ldots, \boldsymbol{y}^{(l)} \in \mathbb{R}^{1 \times d}$
**Input:** A vector with all zeros $\boldsymbol{z} \in \mathbb{R}^{1 \times (k+l)}$
**Output:** Similarity score $DMJ$
    $\boldsymbol{X} \leftarrow$ STACK_ROWS$(\boldsymbol{x}^{(1)}, \boldsymbol{x}^{(2)} \ldots, \boldsymbol{x}^{(k)})$
    $\boldsymbol{Y} \leftarrow$ STACK_ROWS$(\boldsymbol{y}^{(1)}, \boldsymbol{y}^{(2)} \ldots, \boldsymbol{y}^{(l)})$
    $\boldsymbol{U} \leftarrow$ STACK_ROWS$(\boldsymbol{X}, \boldsymbol{Y})$
    $\boldsymbol{x} \leftarrow$ MAX_POOL_ELEMENTWISE$(\boldsymbol{x}^{(1)}\boldsymbol{U}^T, \boldsymbol{x}^{(2)}\boldsymbol{U}^T \ldots, \boldsymbol{x}^{(k)}\boldsymbol{U}^T, \boldsymbol{z})$
    $\boldsymbol{y} \leftarrow$ MAX_POOL_ELEMENTWISE$(\boldsymbol{y}^{(1)}\boldsymbol{U}^T, \boldsymbol{y}^{(2)}\boldsymbol{U}^T \ldots, \boldsymbol{y}^{(l)}\boldsymbol{U}^T, \boldsymbol{z})$

    $\boldsymbol{r} \leftarrow$ MIN_POOL_ELEMENTWISE$(\boldsymbol{x}, \boldsymbol{y})$
    $\boldsymbol{q} \leftarrow$ MAX_POOL_ELEMENTWISE$(\boldsymbol{x}, \boldsymbol{y})$
    DMJ $\leftarrow \sum_{i=1:(k+l)} \boldsymbol{r}_i / \sum_{i=1:(k+l)} \boldsymbol{q}_i$

---

$$\boldsymbol{\mu}^{(i)} = \boldsymbol{e}^{(i)} \boldsymbol{W} \boldsymbol{U}^T. \tag{1}$$

Here $\boldsymbol{W} \in \mathbb{R}^{N \times d}$ is the word embedding matrix and $\boldsymbol{U} \in \mathbb{R}^{K \times d}$ is the 'universe' matrix. Let us dissect the above expression. First, we convert a one-hot vector into a word embedding $\boldsymbol{w}^{(i)} = \boldsymbol{e}^{(i)} \boldsymbol{W}$. This is just an embedding lookup and is exactly the same as the embedding layer in neural networks. Next, we compute a vector of similarities $\boldsymbol{\mu}^{(i)} = \boldsymbol{w}^{(i)} \boldsymbol{U}^T$ between $\boldsymbol{w}^{(i)}$ and all the $K$ vectors in the universe. The most sensible choice for the universe matrix is the word embedding matrix itself, i.e. $\boldsymbol{U} = \boldsymbol{W}$. In that case, the membership vector $\boldsymbol{\mu}^{(i)}$ has the same dimensionality as $\boldsymbol{e}^{(i)}$ but contains similarities between the word $w_i$ and every word in the vocabulary (including itself).

The second step is to combine all $\boldsymbol{\mu}^{(i)}$ back into a sentence membership vector $\boldsymbol{\mu}^s$. At this point, it's very tempting to just sum or average over all $\boldsymbol{\mu}^{(i)}$, i.e. compute $\frac{1}{N} \sum_{i=1}^N c_i \boldsymbol{\mu}^{(i)}$. But we remember: in fuzzy set theory the union of the membership vectors is realised by the element-wise max-pooling. In other words, we don't take the average but max-pool instead:

$$\boldsymbol{\mu}^S = \max_{i=1}^N c_i \boldsymbol{\mu}^{(i)}. \tag{2}$$

Here the max returns a vector where each dimension contains the maximum value along that dimension across all $N$ input vectors. In NLP this is also known as max-over-time pooling (Collobert et al., 2011). Note that any given sentence $S$ usually contains only a small portion of the total vocabulary and so most word counts $c_i$ will be 0. If the count $c_i$ is 0, then we have no need for $\boldsymbol{\mu}^{(i)}$ and can avoid a lot of useless computations, though we must remember to include the zero vector in the max-pooling operation.

We call the sentence membership vector $\boldsymbol{\mu}^S$ the fuzzy bag-of-words (FBoW) and the procedure that converts classical BoW $\boldsymbol{b}^S$ into fuzzy BoW $\boldsymbol{\mu}^S$ the 'sentence fuzzification'.

### 2.2.1 THE FUZZY JACCARD INDEX

Suppose we have two fuzzy BoW $\boldsymbol{\mu}^A$ and $\boldsymbol{\mu}^B$. How can we compare them? Since FBoW are just vectors, we can use the standard cosine similarity $\cos(\boldsymbol{\mu}^A, \boldsymbol{\mu}^B)$. On the other hand, FBoW are also fuzzy sets and as such can be compared via fuzzy SSMs. We simply copy the definitions of fuzzy union, intersection and cardinality from Section 2.1 and write down the fuzzy Jaccard index:

$$\text{Jaccard}(A, B) = \frac{|A \cap B|}{|A \cup B|} \qquad \xrightarrow{\text{fuzzy}} \qquad \text{FJaccard}(\boldsymbol{\mu}^A, \boldsymbol{\mu}^B) = \frac{\sum_{i=1}^K \min(\boldsymbol{\mu}_i^A, \boldsymbol{\mu}_i^B)}{\sum_{i=1}^K \max(\boldsymbol{\mu}_i^A, \boldsymbol{\mu}_i^B)}.$$

Exactly the same can be repeated for other SSMs. In practice we found their performance to be almost equivalent but always better than standard cosine similarity (see Appendix B).

### 2.2.2 SMALLER UNIVERSES AND MAX-POOLED WORD VECTORS

So far we considered the universe and the word embedding matrix to be the same, i.e. $\boldsymbol{U} = \boldsymbol{W}$. This means any FBoW $\boldsymbol{\mu}^S$ contains similarities to all the words in the vocabulary and has exactly the same dimensionality as the original BoW $\boldsymbol{b}^S$. Unlike BoW, however, FBoW is almost never sparse. This motivates us to choose the matrix $\boldsymbol{U}$ with fewer rows that $\boldsymbol{W}$. For example, the top principal axes of $\boldsymbol{W}$ could work. Alternatively, we could cluster $\boldsymbol{W}$ into $k$ clusters and keep the centroids. Of course, the rows of such $\boldsymbol{U}$ are no longer word vectors but instead some abstract entities.

A more radical but completely non-parametric solution is to choose $\boldsymbol{U} = \boldsymbol{I}$, where $\boldsymbol{I} \in \mathbb{R}^{d \times d}$ is just the identity matrix. Then the word fuzzifier reduces to a word embedding lookup:

$$\boldsymbol{\mu}^{(i)} = \boldsymbol{e}^{(i)} \boldsymbol{W} \boldsymbol{U}^T = \boldsymbol{e}^{(i)} \boldsymbol{W} \boldsymbol{I}^T = \boldsymbol{e}^{(i)} \boldsymbol{W} = \boldsymbol{w}^{(i)}. \tag{3}$$

The sentence fuzzifier then simply max-pools all the word embeddings found in the sentence:

$$\boldsymbol{\mu}^S = \max_{w_i \in S} c_i \boldsymbol{w}^{(i)}. \tag{4}$$

From this we see that max-pooled word vectors are only a special case of fuzzy BoW. Remarkably, when word vectors are trained unsupervised, this simple representation combined with the fuzzy Jaccard index is already a stronger baseline for semantic textual similarity than the averaged word vector with cosine similarity, as we will see in Section 4.

More importantly, the fuzzy Jaccard index works for max-pooled word vectors but completely fails for averaged word vectors. This empirically validates the connection between fuzzy BoW representations and the max-pooling operation described above.

### 2.2.3 THE DYNAMAX ALGORITHM

From the linear-algebraic point of view, fuzzy BoW is really the same as projecting word embeddings on a subspace of $\mathbb{R}^d$ spanned by the rows of $\boldsymbol{U}$, followed by max-pooling of the features extracted by this projection. A fair question then is the following. If we want to compare two sentences, what subspace should we project on? It turns out that if we take word embeddings for the first sentence and the second sentence and stack them into matrix $\boldsymbol{U}$, this seems to be a sufficient space to extract all the features needed for semantic similarity. We noticed this empirically, and while some other choices of $\boldsymbol{U}$ do give better results, finding a principled way to construct them remains future work. The matrix $\boldsymbol{U}$ is not static any more but instead changes dynamically depending on the sentence pair. We call this approach Dynamic Max or DynaMax and provide pseudocode in Algorithm 1.

### 2.2.4 PRACTICAL CONSIDERATIONS

Just as SoW is a special case of BoW, we can build the fuzzy set-of-words (FSoW) where the word counts $c_i$ are binary. The performance of FSoW and FBoW is comparable, with FBoW being marginally better. For simplicity, we implement FSoW in Algorithm 1 and in all our experiments.

As evident from Equation (1), we use dot product as opposed to (scaled or clipped) cosine similarity for the membership functions. This is a reasonable choice as most unsupervised and some supervised word vectors maximise dot products in their objectives. For further analysis, see Appendix A.

## 3 RELATED WORK

Any method that casts semantic similarity between sentences as the proximity of their vector representations is related to our work. Among those, the ones that strengthen bag-of-words by incorporating the sense of similarity between individual words are the most relevant.

The standard Vector Space Model (VSM) basis $\boldsymbol{e}^{(i)}$ is orthonormal and so the BoW model treats all words as equally different. Sidorov et al. (2014) proposed the 'soft cosine measure' to alleviate this issue. They build a non-orthogonal basis $\boldsymbol{f}^{(i)}$ where $\cos(\boldsymbol{f}^{(i)}, \boldsymbol{f}^{(j)}) = \text{sim}(w_i, w_j)$, i.e. the cosine similarity between the basis vectors is given by similarity between words. Next, they rewrite BoW in

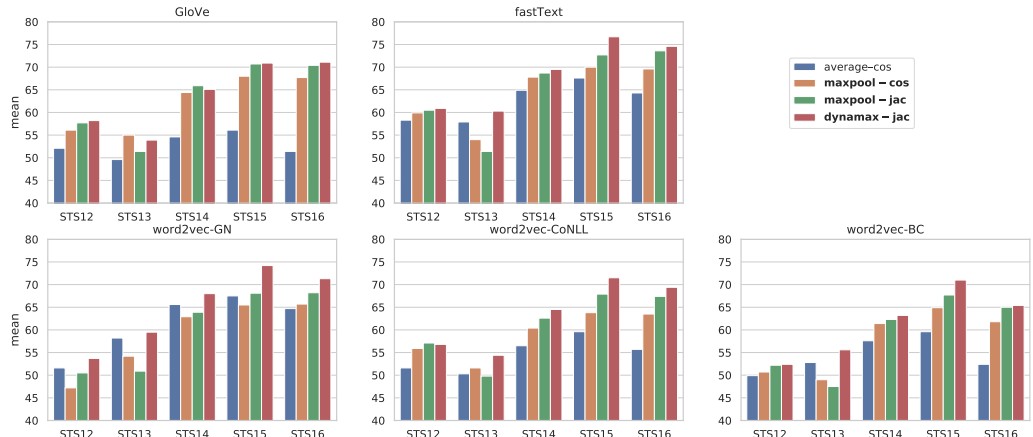

Figure 1: Each plot shows the mean Pearson correlation on STS tasks for a different flavour of word vector, comparing different combinations of fuzzy BoW representation (either averaged or max-pooled, or the DynaMax approach) and similarity measure (either cosine or Jaccard). The bolded methods are ones proposed in the present work. Note that averaged vectors with Jaccard similarity are not included in these plots, as they consistently perform 20-50 points worse than other methods; this is predicted by our analysis as averaging is not an appropriate union operation in fuzzy set theory. In virtually every case, max-pooled with cosine outperforms averaged with cosine, which is in turn outperformed by max-pooled and DynaMax with Jaccard. An exception to the trend is STS13, for which the SMT subtask dataset is no longer publicly available; this may have impacted the performance when averaged over different types of subtasks.

terms of $\boldsymbol{f}^{(i)}$ and compute cosine similarity between transformed representations. However, when $\cos(\boldsymbol{f}^{(i)}, \boldsymbol{f}^{(j)}) = \cos(\boldsymbol{w}_i, \boldsymbol{w}_j)$, where $\boldsymbol{w}_i, \boldsymbol{w}_j$ are word embeddings, their approach is equivalent to cosine similarity between averaged word embeddings, i.e. the standard baseline.

Kusner et al. (2015) consider L1-normalised bags-of-words (nBoW) and view them as a probability distributions over words. They propose the Word Mover's Distance (WMD) as a special case of the Earth Mover's Distance (EMD) between nBoW with the cost matrix given by pairwise Euclidean distances between word embeddings. As such, WMD does not build any new representations but puts a lot of structure into the distance between BoW.

Zhao & Mao (2017) proposed an alternative version of fuzzy BoW that is conceptually similar to ours but executed very differently. They use clipped cosine similarity between word embeddings to compute the membership values in the word fuzzification step. We use dot product not only because it is theoretically more general but also because dot product leads to significant improvements on the benchmarks. More importantly, however, their sentence fuzzification step uses sum to aggregate word membership vectors into a sentence membership vector. We argue that max-pooling is a better choice because it corresponds to the fuzzy union. Had we used the sum, the representation would have really reduced to a (projected) summed word vector. Lastly, they use FBoW as features for a supervised model but stop short of considering any fuzzy similarity measures, such as fuzzy Jaccard index.

Jimenez et al. (2010; 2012; 2013; 2014; 2015) proposed and developed soft cardinality as a generalisation to the classical set cardinality. In their framework set membership is crisp, just as in classical set theory. However, once the words are in a set, their contribution to the overall cardinality depends on how similar they are to each other. The intuition is that the set $A = \{$'lion', 'tiger', 'leopard'$\}$ should have cardinality much less than 3, because $A$ contains very similar elements. Likewise, the set $B = \{$'lion', 'airplane', 'carrot'$\}$ deserves a cardinality closer to 3. We see that the soft cardinality framework is very different from our approach, as it 'does not consider uncertainty in the membership of a particular element; only uncertainty as to the contribution of an element to the cardinality of the set' (Jimenez et al., 2010).

## 4 EXPERIMENTS

To evaluate the proposed similarity measures we set up a series of experiments on the established STS tasks, part of the SemEval shared task series 2012-2016 (Agirre et al., 2012; 2013; 2014; Agirre, 2015; Agirre et al., 2016; Cer et al., 2017). The idea behind the STS benchmarks is to measure

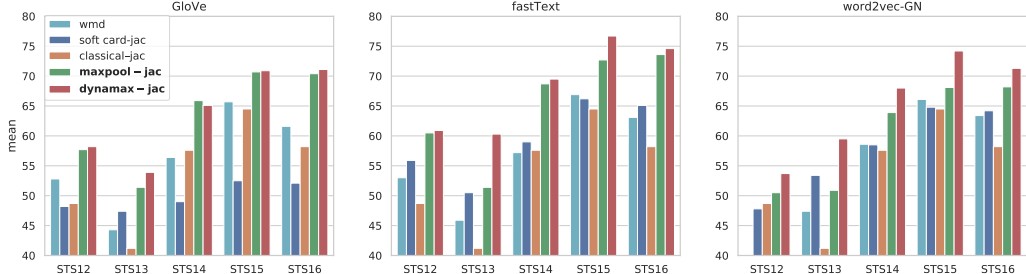

Figure 2: Each plot shows the mean Pearson correlation on STS tasks for a different flavour of word vector, comparing other BoW-based methods to ones using fuzzy Jaccard similarity. The bolded methods are ones proposed in the present work. We observe that even classical crisp Jaccard is a fairly reasonable baseline, but it is greatly improved by the fuzzy set treatment. Both max-pooled word vectors with Jaccard and DynaMax outperform the other methods by a comfortable margin, and the max-pooled version in particular performs astonishingly well given its great simplicity.

how well the semantic similarity scores computed by a system (algorithm) correlate with human judgements. Each year's STS task itself consists of several subtasks. By convention, we report the mean Pearson correlation between system and human scores, where the mean is taken across all the subtasks in a given year.

Our implementation wraps the SentEval toolkit (Conneau & Kiela, 2018) and is available on GitHub[1]. We also rely on the following publicly available word embeddings: GloVe (Pennington et al., 2014) trained on Common Crawl (840B tokens); fastText (Bojanowski et al., 2016) trained on Common Crawl (600B tokens); word2vec (Mikolov et al., 2013b;c) trained on Google News, CoNLL (Zeman et al., 2017), and Book Corpus (Zhu et al., 2015); and several types of supervised paraphrastic vectors – PSL (Wieting et al., 2015), PP-XXL (Wieting et al., 2016), and PNMT (Wieting & Gimpel, 2018).

We estimated word frequencies on an English Wikipedia dump dated July 1[st] 2017 and calculated word weights using the same approach and parameters as in Arora et al. (2017). Note that these weights can in fact be derived from word vectors and frequencies alone rather than being inferred from the validation set (Ethayarajh, 2018), making our techniques fully unsupervised. Finally, as the STS'13 SMT dataset is no longer publicly available, the mean Pearson correlations reported in our experiments involving this task have been re-calculated accordingly.

We first ran a set of experiments validating the insights and derivations described in Section 2. These results are presented in Figure 1. The main takeaways are the following:

- Max-pooled word vectors outperform averaged word vectors in most tasks.
- Max-pooled vectors with cosine similarity perform worse than max-pooled vectors with fuzzy Jaccard similarity. This supports our derivation of max-pooled vectors as a special case of fuzzy BoW, which thus should be compared via fuzzy set similarity measures and not cosine similarity (which would be an arbitrary choice).
- Averaged vectors with fuzzy Jaccard similarity completely fail. This is because fuzzy set theory tells us that the average is not a valid fuzzy union operation, so a fuzzy set similarity is not appropriate for this representation.
- DynaMax shows the best performance across all tasks, possibly thanks to its superior ability to extract and max-pool good features from word vectors.

Next we ran experiments against some of the related methods described in Section 3, namely WMD (Kusner et al., 2015) and soft cardinality (Jimenez et al., 2015) with clipped cosine similarity as an affinity function and the softness parameter $p = 1$. From Figure 2, we see that even classical Jaccard index is a reasonable baseline, but fuzzy Jaccard especially in the DynaMax formulation handily outperforms comparable methods.

For context and completeness, we also compare against other popular sentence representations from the literature in Table 1. We include the following methods: BoW with ELMo embeddings

---

[1]https://github.com/Babylonpartners/fuzzymax

Table 1: Mean Pearson correlation on STS tasks for a variety of methods in the literature. Bolded values indicate best results per task and word vector where applicable, while underlined values indicate overall best per task. All previous results are taken from Perone et al. (2018) (only two significant figures provided) and Subramanian et al. (2018b). Note that avg-cos refers to taking the average word vector and comparing by cosine similarity, and word2vec refers to the Google News version. Clearly more sophisticated methods of computing sentence representations do not shine on the unsupervised STS tasks when compared to these simple BoW methods with high-quality word vectors and the appropriate similarity metric. † indicates the only STS13 result (to our knowledge) that includes the SMT subtask.

| Approach | STS12 | STS13 | STS14 | STS15 | STS16 |
|---|---|---|---|---|---|
| ELMo (BoW) | 55 | 53 | 63 | 68 | 60 |
| Skip-Thought | 41 | 29 | 40 | 46 | 52 |
| InferSent | 61 | 56 | 68 | 71 | 71 |
| USE (DAN) | 59 | 59 | 68 | 72 | 70 |
| USE (Transformer) | 61 | 64 | 71 | 74 | 74 |
| STN (multitask) | 60.6 | 54.7† | 65.8 | 74.2 | 66.4 |
| GloVe avg-cos | 52.1 | 49.6 | 54.6 | 56.1 | 51.4 |
| GloVe DynaMax | **58.2** | **53.9** | **65.1** | **70.9** | **71.1** |
| fastText avg-cos | 58.3 | 57.9 | 64.9 | 67.6 | 64.3 |
| fastText DynaMax | **60.9** | **60.3** | **69.5** | **76.7** | **74.6** |
| word2vec avg-cos | 51.6 | 58.2 | 65.6 | 67.5 | 64.7 |
| word2vec DynaMax | **53.7** | **59.5** | **68.0** | **74.2** | **71.3** |
| PSL avg-cos | 52.7 | 51.8 | 59.6 | 61.0 | 54.1 |
| PSL DynaMax | **58.2** | **54.3** | **66.2** | **72.4** | **66.5** |
| PP-XXL avg-cos | 61.3 | **65.6** | **72.7** | 77.0 | **71.1** |
| PP-XXL DynaMax | **63.6** | 62.2 | **72.7** | **77.9** | 70.8 |
| PNMT avg-cos | 65.6 | **68.9** | **76.3** | 79.4 | **77.2** |
| PNMT DynaMax | **66.0** | 65.7 | 75.9 | **80.1** | 76.7 |

(Peters et al., 2018), Skip-Thought (Kiros et al., 2015), InferSent (Conneau et al., 2017), Universal Sentence Encoder with DAN and Transformer (Cer et al., 2018b), and STN multitask embeddings (Subramanian et al., 2018b). These experiments lead to an interesting observation:

- PNMT embeddings are the current state-of-the-art on STS tasks. PP-XXL and PNMT were trained supervised to directly optimise cosine similarity between average word vectors on very large paraphrastic datasets. By contrast, DynaMax is completely unrelated to the training objective of these vectors, yet has an equivalent performance.

Finally, another well-known and high-performing simple baseline was proposed by Arora et al. (2017). However, as also noted by Mu & Viswanath (2018), this method is still offline because it computes the sentence embeddings for the entire dataset, then performs PCA and removes the top principal component. While their method makes more assumptions than ours, nonetheless we make a head-to-head comparison with them in Table 2 using the same word vectors as in Arora et al. (2017), showing that DynaMax is still quite competitive.

To strengthen our empirical findings, we provide ablation studies for DynaMax in Appendix C, showing that the different components of the algorithm each contribute to its strong performance. We also conduct significance testing in Appendix D by constructing bias-corrected and accelerated (BCa) bootstrap confidence intervals (Efron, 1987) for the delta in performance between two algorithms. This constitutes, to the best of our knowledge, the first attempt to study statistical significance on the STS benchmarks with this type of non-parametric analysis that respects the statistical peculiarities of these datasets.

Table 2: Mean Pearson correlations on STS tasks comparing DynaMax against Arora et al. (2017)'s avg-SIF+PCA method. Bolded values indicate best results per task and word vectors (both methods can be applied to any word vectors). All methods use SIF word weights as described in Arora et al. (2017); in this case average word vector with cosine similarity (avg-SIF in the table) is equivalent to the Arora et al. (2017)'s method without PCA, so we include it for additional context. Note that removing the first principal component requires computation on the entire test set (see Algorithm 1 in Arora et al. (2017)), whereas DynaMax is completely independent of the test set. Even with this distinction, avg-SIF+PCA and DynaMax perform comparably, and both generally outperform avg-SIF although use of PSL vectors closes the gap considerably.

| Vectors | Similarity | STS12 | STS13 | STS14 | STS15 | STS16 |
|---|---|---|---|---|---|---|
| | avg-SIF | 59.2 | 59.9 | 62.9 | 62.8 | 63.0 |
| GloVe | avg-SIF+PCA | 58.5 | **65.5** | **69.3** | 70.2 | 69.6 |
| | DynaMax-SIF | **61.1** | 61.5 | **69.3** | **73.1** | **71.7** |
| | avg-SIF | 61.5 | 66.7 | 71.5 | 72.8 | 69.7 |
| PSL | avg-SIF+PCA | 61.0 | **67.8** | **72.9** | 75.8 | 71.9 |
| | DynaMax-SIF | **63.2** | 64.8 | 72.8 | **77.6** | **73.3** |

## 5 CONCLUSION

In this work we combine word embeddings with classic BoW representations using fuzzy set theory. We show that max-pooled word vectors are a special case of FBoW, which implies that they should be compared via the fuzzy Jaccard index rather than the more standard cosine similarity. We also present a simple and novel algorithm, DynaMax, which corresponds to projecting word vectors onto a subspace dynamically generated by the given sentences before max-pooling over the features. DynaMax outperforms averaged word vectors compared with cosine similarity on every benchmark STS task when word vectors are trained unsupervised. It even performs comparably to supervised vectors that directly optimise cosine similarity between paraphrases, despite being completely unrelated to that objective.

Both max-pooled vectors and DynaMax constitute strong baselines for further studies in the area of sentence representations. Yet, these methods are not limited to NLP and word embeddings, but can in fact be used in any setting where one needs to compute similarity between sets of elements that have rich vector representations. We hope to have demonstrated the benefits of experimenting more with similarity metrics based on the building blocks of meaning such as words, rather than complex representations of the final objects such as sentences.

### ACKNOWLEDGMENTS

We would like to thank John Wieting for sharing with us his latest state-of-the-art ParaNMT embeddings, so that we could include the most up-to-date comparisons in the present work.

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

Table 3: Mean Pearson correlation on STS tasks for DynaMax and averaged word vector, all using normalised word vectors. Bolded values indicate best results per task and word vector type. Note that DynaMax still outperforms avg-cos across the board, but both approaches lose to their unnormalised counterparts as reported in Table 1.

| Vectors | Approach | STS12 | STS13 | STS14 | STS15 | STS16 |
|---------|----------|-------|-------|-------|-------|-------|
| **GloVe** | avg-cos | 47.1 | 44.9 | 49.7 | 51.9 | 44.0 |
| | DynaMax | **53.7** | **47.8** | **59.5** | **66.3** | **62.9** |
| **fastText** | avg-cos | 47.6 | 46.1 | 54.5 | 58.8 | 49.6 |
| | DynaMax | **51.6** | **46.3** | **59.6** | **68.5** | **62.8** |
| **word2vec** | avg-cos | 45.2 | 49.3 | 57.3 | 59.1 | 51.8 |
| | DynaMax | **47.6** | **49.7** | **60.7** | **68.0** | **62.8** |

## A  Normalised Vectors and $[0, 1]$-fuzzy Sets

In the word fuzzification step the membership values for a word $w$ are obtained through a similarity function $\text{sim}(\boldsymbol{w}, \boldsymbol{u}^{(j)})$ between the word embedding $\boldsymbol{w}$ and the rows of the universe matrix $\boldsymbol{U}$, i.e.

$$\boldsymbol{\mu} = [\text{sim}(\boldsymbol{w}, \boldsymbol{u}^{(1)}), \text{sim}(\boldsymbol{w}, \boldsymbol{u}^{(2)}), \dots, \text{sim}(\boldsymbol{w}, \boldsymbol{u}^{(K)})].$$

In Section 2.2, $\text{sim}(\boldsymbol{w}, \boldsymbol{u}^{(j)})$ was the dot product $\boldsymbol{w} \cdot \boldsymbol{u}^{(j)}$ and we could simply write $\boldsymbol{\mu} = \boldsymbol{w}\boldsymbol{U}^T$. There are several reasons why we chose a similarity function that takes values in $\mathbb{R}$ as opposed to $[0, 1]$.

First, we can always map the membership values from $\mathbb{R}$ to $(0, 1)$ and vice versa using, e.g. the logistic function $\sigma(x) = \frac{1}{1+e^{-ax}}$ with an appropriate scaling factor $a > 0$. Intuitively, large negative membership values would imply the element is really not in the set and large positive values mean it is really in the set. Of course, here both 'large' and 'really' depend on the scaling factor $a$. In any case, we see that the choice of $\mathbb{R}$ vs. $[0, 1]$ is not very important mathematically. Interestingly, since we always max-pool with a zero vector, fuzzy BoW will not contain any negative membership values. This was not our intention, just a by-product of the model.

Secondly, note that the membership function for multisets takes values in $\mathbb{N}^{\geq 0}$, i.e. the nonnegative integers. These values are already outside $[0, 1]$ and we see that the standard $[0, 1]$-fuzzy sets are incompatible with multisets. On the other hand, a membership function that takes values in $\mathbb{R}$ can directly model sets, multisets, fuzzy sets, and fuzzy multisets.

For completeness, let us insist on the range $[0, 1]$ and choose $\text{sim}(\boldsymbol{w}, \boldsymbol{u}^{(j)})$ to be the clipped cosine similarity $\max(0, \cos(\boldsymbol{w}, \boldsymbol{u}^{(j)}))$. This is in fact equivalent to simply normalising the word vectors. Indeed, the dot product and cosine similarity become the same after normalisation, and max-pooling with the zero vector removes all the negative values, so the resulting representation is guaranteed to be a $[0, 1]$-fuzzy set. Our results for normalised word vectors are presented in Table 3.

After comparing Tables 1 and 3 we can draw two conclusions. Namely, DynaMax still outperforms avg-cos by a large margin even when word vectors are normalised. However, normalisation hurts both approaches and should generally be avoided. This is not surprising since the length of word vectors is correlated with word importance, so normalisation essentially makes all words equally important (Schakel & Wilson, 2015).

## B  Comparison of Fuzzy Set Similarity Measures

In Section 2 we mentioned several set similarity measures such as Jaccard (Jaccard, 1901), Otsuka-Ochiai (Otsuka, 1936; Ochiai, 1957) and Sørensen–Dice (Dice, 1945; Sørensen, 1948) coefficients. Here in Table 4, we show that fuzzy versions of the above coefficients have almost identical performance, thus confirming that our results are in no way specific to the Jaccard index.

Table 4: Mean Pearson correlation on STS tasks for different fuzzy SSMs. The performance is almost identical across the board.

| Vectors | SSM | STS12 | STS13 | STS14 | STS15 | STS16 |
|---------|-----|-------|-------|-------|-------|-------|
| **GloVe** | Jaccard | 58.2 | 53.9 | 65.1 | 70.9 | 71.1 |
| | Otsuka | 58.3 | 53.4 | 65.2 | 70.3 | 70.5 |
| | Dice | 58.5 | 53.2 | 64.9 | 70.1 | 70.4 |
| **fastText** | Jaccard | 60.9 | 60.3 | 69.5 | 76.7 | 74.6 |
| | Otsuka | 61.0 | 60.1 | 69.7 | 76.1 | 74.0 |
| | Dice | 61.3 | 59.5 | 69.4 | 76.0 | 73.8 |
| **word2vec** | Jaccard | 53.7 | 59.5 | 68.0 | 74.2 | 71.3 |
| | Otsuka | 51.5 | 58.8 | 67.7 | 73.4 | 70.1 |
| | Dice | 51.9 | 58.7 | 67.5 | 73.3 | 70.0 |

Table 5: Mean Pearson correlation on STS tasks for the ablation studies. As described in Appendix C, it is clear that the three components of the algorithm — the dynamic universe, the max-pooling operation, and the fuzzy Jaccard index — all contribute to the strong performance of DynaMax-Jaccard.

| Ablation on | Approach | STS12 | STS13 | STS14 | STS15 | STS16 |
|-------------|----------|-------|-------|-------|-------|-------|
| | DynaMax Jaccard | **60.9** | 60.3 | **69.5** | **76.7** | **74.6** |
| **Universe** | Max Jaccard | 60.5 | 51.4 | 68.7 | 72.7 | 73.6 |
| | RandomMax Jaccard | 58.6 | 52.2 | 67.0 | 72.2 | 71.3 |
| **Similarity** | DynaMax cosine | 60.2 | **62.2** | 68.1 | 74.2 | 69.7 |
| **Pooling Op.** | DynaAvg Jaccard | 52.1 | 45.8 | 52.0 | 60.5 | 54.9 |
| | DynaSum Jaccard | 47.8 | 34.6 | 38.7 | 45.7 | 41.1 |
| | DynaMin Jaccard | 28.4 | 21.5 | 27.1 | 34.4 | 37.2 |
| **Pool & Sim.** | DynaAvg cosine | 55.6 | 53.4 | 56.4 | 58.1 | 50.7 |

## C DynaMax Ablation Studies

The DynaMax-Jaccard similarity (Algorithm 1) consists of three components: the dynamic universe, the max-pooling operation, and the fuzzy Jaccard index. As with any algorithm, it is very important to track the sources of improvements. Consequently, we perform a series of ablation studies in order to isolate the contribution of each component. For brevity, we focus on fastText because it produced the strongest results for both the DynaMax and the baseline (Figure 1).

The results of the ablation study are presented in Table 5. First, we show that the dynamic universe is superior to other sensible choices, such as the identity and random $300 \times 300$ projection with components drawn from $\mathcal{N}(0, 1)$. Next, we show that the fuzzy Jaccard index beats the standard cosine similarity on 4 out 5 benchmarks. Finally, we find that max considerably outperforms other pooling operations such as averaging, sum and min. We conclude that all three components of DynaMax are very important. It is clear that max-pooling is the top contributing factor, followed by the dynamic universe and the fuzzy Jaccard index, whose contributions are roughly equal.

## D Significance Analysis

As discussed in Section 4, the core idea behind the STS benchmarks is to measure how well the semantic similarity scores computed by a system (algorithm) correlate with human judgements. In this section we provide detailed results and significance analysis for all 24 STS subtasks. Our approach can be formally summarised as follows. We assume that the human scores $H$, the system scores $A$ and the baseline system scores $B$ jointly come from some trivariate distribution $P(H, A, B)$, which is specific to each subtask. To compare the performance of two systems, we compute the sample Pearson correlation coefficients $r_{AH}$ and $r_{BH}$. Since these correlations share the variable $H$, they

are themselves dependent. There are several parametric tests for the difference between dependent correlations; however, their appropriateness beyond the assumptions of normality remains an active area of research (Hittner et al., 2003; Wilcox & Tian, 2008; Wilcox, 2009). The distributions of the human scores in the STS tasks are generally not normal; what's more, they vary greatly depending on the subtask (some are multimodal, others are skewed, etc.).

Fortunately, nonparametric resampling-based approaches, such as bootstrap (Efron & Tibshirani, 1994), present an attractive alternative to parametric tests when the distribution of the test statistic is unknown. In our case, the statistic is simply the difference between two correlations $\hat{\Delta} = r_{AH} - r_{BH}$. The main idea behind bootstrap is intuitive and elegant: just like a sample is drawn from the population, a large number of 'bootstrap' samples can be drawn from the actual sample. In our case, the dataset consists of triplets $\mathcal{D} = \{(h_i, a_i, b_i)\}_{i=1}^{M}$. Each bootstrap sample is a result of drawing $M$ data points from $\mathcal{D}$ with replacement. Finally, we approximate the distribution of $\Delta$ by evaluating it on a large number of bootstrap samples, in our case ten thousand. We use this information to construct bias-corrected and accelerated (BCa) 95% confidence intervals for $\Delta$. BCa (Efron, 1987) is a fairly advanced second-order method that accounts for bias and skewness in the bootstrapped distributions, effects we did observe to a small degree in certain subtasks.

Once we have the confidence interval for $\Delta$, the decision rule is then simple: if zero is inside the interval, then the difference between correlations is not significant. Inversely, if zero is outside, we may conclude that the two approaches lead to statistically different results. The location of the interval further tells us which one performs better. The results are presented in Table 6. In summary, out of 72 experiments we significantly outperform the baseline in 56 (77.8%) and underperform in only one (1.39%), while in the remaining 15 (20.8%) the differences are nonsignificant. We hope our analysis is useful to the community and will serve as a good starting point for conducting thorough significance testing on the current as well as future STS benchmarks.

Table 6: Pearson correlation for DynaMax and averaged word vector, and the 95% confidence interval for the difference between two correlations. According to this analysis, DynaMax significantly outperforms the baseline in 77.8% of the experiments and significantly loses in only 1.39% of them. In the remaining 20.8% of the experiments the two approaches perform statistically the same.

| | | GloVe | | | fastText | | | word2vec | |
|---|---|---|---|---|---|---|---|---|---|
| | | DynaMax-J | Avg. Cos. | Δ95% CI | DynaMax-J | Avg. Cos. | Δ95% CI | DynaMax-J | Avg. Cos. | Δ95% CI |
| STS12 | MSRpar | **49.41** | 42.55 | [3.20, 10.67] | **48.94** | 40.39 | [4.78, 12.35] | **41.74** | **39.72** | [-1.03, 4.99] |
| | MSRvid | **71.92** | 66.21 | [3.97, 7.70] | **76.20** | 73.77 | [1.13, 3.78] | 76.86 | **78.11** | [-2.25, -0.28] |
| | SMTeuroparl | **48.43** | 48.36 | [-4.60, 5.92] | **53.08** | 53.03 | [-3.07, 3.33] | **28.03** | 16.06 | [8.69, 15.05] |
| | surprise.OnWN | **69.86** | 57.03 | [9.69, 16.44] | **72.79** | 68.92 | [1.84, 6.03] | **71.26** | **71.06** | [-1.39, 1.73] |
| | surprise.SMTnews | **51.47** | 46.27 | [0.03, 10.80] | **53.26** | 55.20 | [-6.12, 2.15] | **50.44** | **52.91** | [-6.38, 1.45] |
| STS13 | FNWN | **39.79** | **38.21** | [-6.38, 9.99] | **42.34** | **39.83** | [-5.98, 10.72] | **42.34** | **41.22** | [-6.93, 8.40] |
| | headlines | **69.91** | 63.39 | [4.81, 8.33] | **73.13** | 70.83 | [1.04, 3.61] | **66.66** | 65.22 | [0.15, 2.74] |
| | OnWN | **52.12** | 47.20 | [2.00, 8.06] | **65.35** | 63.03 | [0.33, 4.36] | **69.36** | 68.29 | [-0.63, 2.71] |
| STS14 | deft-forum | **43.29** | 30.02 | [8.17, 18.94] | **47.16** | 40.19 | [3.17, 11.03] | **47.27** | 42.66 | [0.86, 9.32] |
| | deft-news | **70.55** | 64.95 | [0.98, 10.93] | **71.04** | **71.15** | [-3.73, 3.34] | 65.84 | **67.28** | [-4.66, 2.03] |
| | headlines | **64.49** | 58.67 | [3.92, 7.98] | **68.22** | 66.03 | [0.99, 3.54] | **63.66** | 61.88 | [0.51, 3.32] |
| | images | **75.05** | 62.38 | [10.11, 15.70] | **79.39** | 71.45 | [6.15, 10.03] | **80.51** | 77.46 | [1.84, 4.35] |
| | OnWN | **63.00** | 57.71 | [3.10, 7.77] | **72.83** | 70.47 | [0.92, 3.84] | **75.43** | **75.12** | [-0.83, 1.45] |
| | tweet-news | **74.30** | 53.87 | [16.44, 25.51] | **78.41** | 70.18 | [5.78, 11.56] | **75.47** | 69.26 | [4.04, 9.07] |
| STS15 | answers-forums | **61.94** | 36.66 | [20.01, 30.94] | **73.57** | 56.91 | [12.51, 21.44] | **66.44** | 53.95 | [8.10, 17.23] |
| | answers-students | **73.53** | 63.62 | [7.34, 13.49] | **75.82** | 71.81 | [2.45, 5.59] | **75.07** | 72.78 | [0.96, 3.79] |
| | belief | **67.21** | 44.78 | [17.48, 29.42] | **76.14** | 60.62 | [11.61, 21.38] | **75.83** | 61.89 | [10.45, 18.08] |
| | headlines | **72.26** | 66.21 | [4.20, 8.20] | **74.45** | 72.53 | [0.84, 3.06] | **69.95** | 68.72 | [0.23, 2.32] |
| | images | **79.30** | 69.09 | [8.23, 12.56] | **83.33** | 76.12 | [5.65, 8.98] | **83.80** | 80.22 | [2.45, 4.85] |
| STS16 | answer-answer | **59.72** | 40.12 | [13.35, 26.62] | **63.30** | 45.13 | [12.40, 24.85] | **58.78** | 43.14 | [10.26, 21.60] |
| | headlines | **71.71** | 61.38 | [6.88, 14.77] | **73.40** | 70.37 | [1.04, 5.27] | **68.18** | 66.64 | [-0.21, 3.38] |
| | plagiarism | **79.92** | 54.61 | [18.76, 33.16] | **82.68** | 74.49 | [4.07, 13.14] | **82.05** | 76.46 | [2.41, 9.09] |
| | postediting | **80.48** | 53.88 | [21.55, 32.64] | **84.15** | 68.76 | [9.00, 23.95] | **81.73** | 73.35 | [5.52, 12.47] |
| | question-question | **63.51** | 47.21 | [10.06, 25.17] | **69.71** | 62.62 | [2.91, 11.30] | **65.74** | 63.74 | [-1.85, 6.10] |

