# OpenReview forum: "Don't Settle for Average, Go for the Max: Fuzzy Sets and Max-Pooled Word Vectors"
_ICLR.cc/2019/Conference_

### Official Review · AnonReviewer1 · 2018-10-30
**This paper presents (a) a fuzzy bag of word representation and (b) DynaMax, a similarity measure that max pools salient features from sentence pairs.**

**Rating:** 5
**Confidence:** 3

**Review:**

Strengths:
- Good coverage of related work
- Clear presentation of the methods
- Evaluation using established SemEval datasets


Weaknesses:
1. It is not entirely clear what is the connection between fuzzy bag of words and DynaMax. In principle DynaMax can work with other methods too. This point should be elaborated a bit more.
2. It is claimed that the this paper shows that max-pooled word vectors are a special case of fuzzy bag of words. This is not correct. The paper shows how to "convert" one to the other.
3. It is also claimed that point 2 above implies that max-pooled vectors should be compared with the fuzzy Jaccard index instead of cosine similarity. There is no proof or substantial justification to support this.
4. Some relevant work that is missing:
- De Boom, C., Van Canneyt, S., Demeester, T., Dhoedt, B.: Representation learning for very
short texts using weighted word embedding aggregation. Pattern Recognition Letters 80,
150–156 (2016)
- Kenter, T., De Rijke, M.: Short text similarity with word embeddings. In: International on
Conference on Information and Knowledge Management. pp. 1411–1420. ACM (2015)

---

> ### Author Response · Authors · 2018-11-12
> **Thank you!**
>
> Dear Reviewer,
>
> We would like to thank you for your assessment of our paper and positive comments regarding presentation and coverage of related work.
> The Reviewer additionally had some concerns which we would like to address.
>
>
> 1.
> As we explain in Section 2.2.3, the universe of DynaMax contains only the word embeddings from the 2 sentences being compared. If x1, x2,...xk and y1, y2,...,yl are word embeddings for sentences 1 and 2 respectively, then U = [x1; x2;...xk; y1; y2;...;yl]. This construction is also shown in Algorithm 1 (Lines 5-7).
>
> We are not quite sure what the Reviewer meant by "In principle DynaMax can work with other methods too".
>
> The DynaMax-Jaccard (DMJ) similarity has 3 components. The "dynamic" universe U (Section 2.2.3), the max-pooling operation (Eq. 2), and finally the fuzzy Jaccard index (Section 2.2.1). All 3 components are reflected and implemented in Algorithm 1.
> Below we show the change in performance when max-pooling is replaced by average and when fuzzy Jaccard is replaced by cosine similarity.
>
> GloVe                                STS12   STS13   STS14   STS15   STS16
>
> DynaMax Jaccard            58.2      53.9      65.1       70.9      71.1
> DynaMax Cosine             58.2      53.6      63.2       67.2      67.4
>
> DynaAvg. Jaccard            43.5      37.0      38.8       45.3      39.4
> DynaAvg. Cosine	           40.0      39.1      38.3       39.7      31.2
>
> We see that all 3 components in DynaMax-Jaccard are very important. When we replaced Jaccard with cosine, the performance dropped. When we replaced max with average it fell even further. Unfortunately, there is only a limited number of combinations we could report in the paper.
>
>
> 2. > "... max-pooled word vectors are a special case of fuzzy bag of words. This is not correct."
>
> Please allow us to elaborate why max-pooled vectors are in fact a special case of fuzzy BoW:
>
> We deliberately left the matrix U unspecified in the definition of FBoW (Section 2.2, Eq. 1 & 2).
> When U=W, then U represents "concrete" words. We said this was the most intuitive (but not the only) choice.
>
> In some cases, we no longer have concrete words but instead the words are "abstract". We acknowledge this in Section 2.2.2. In case of max-pooled vectors, U is the identity matrix I. However, we can always assign text labels to the rows of I, for example 'dim1', 'dim2', ..., 'dim300'.
> These "words" represent abstract concepts learned by a neural network in its representations. In fact, there has been some work to figure out what these dimensions could contain (e.g. [1])
> More generally, for any vector we can always generate a text label for that vector, and take that as a word.
> The fuzzy BoW is then fuzzy with respect to these abstract words (concepts).
>
> We will clarify this in the updated manuscript.
>
> [1] Yulia Tsvetkov, Manaal Faruqui, and Chris Dyer. Correlation-based Intrinsic Evaluation of Word Vector Representations.
> Proceedings of the 1st Workshop on Evaluating Vector-Space Representations for NLP, pp. 111–115
>
>
> 3. "... max-pooled vectors should be compared with the fuzzy Jaccard index instead of cosine similarity. There is no proof or substantial justification to support this. "
>
> We appreciate there is no proof but we respectfully disagree there is no substantial justification. As discussed above, max-pooled vectors are a special case of fuzzy BoW and so the fuzzy Jaccard index is fully justified for this representation. Empirically fuzzy Jaccard outperforms cosine similarity on most tasks (Figure 1).
>
>
> 4. Thanks for bringing these works to our attention. We're happy to cite them where appropriate.
>
>
> Overall, we showed that word embeddings by themselves (without any weights, tricks or supervision) are still a formidable baseline for semantic textual similarity. We reported up to 20-point increase on standard benchmark datasets. We also tried to rekindle the interest in fuzzy set theory, which is quite underrepresented in the mainstream ML research.
>
> In addition to our replies, we hope the Reviewer can take these contributions into account and perhaps reconsider their score.
>
> Again, thank you very much for your assessment and please do not hesitate to contact us with any further queries.
>
>
> Best wishes,
>
>
>
> ICLR 2019 Conference Paper1058 Authors

---

> ### Author Response · Authors · 2018-11-27
> **Updates to the paper**
>
> Dear Reviewer,
>
>
> We wanted to let you know that the new version of the paper has now been uploaded.
> The main text remains almost the same, however, we have added 4 new sections to the Appendix.
>
> We included ablation studies on the DynaMax in Appendix C; in one of those we compare the dynamic universe, the identity matrix and also the random projection. We hope this further explains the connection between the universe of DynaMax and other universes.
>
> We also hope this addresses the Reviewer's comment saying that "DynaMax can work with other methods too". We show that the best-performing version of DynaMax is the one described in the paper.
>
> Also, we have now cited both works suggested by the Reviewer.
>
>
> Finally, we added many other interesting results, including a novel significance analysis methodology for STS, comparison between different fuzzy similarity coefficients and discussion on [0, 1]-fuzzy sets and their connection with normalised vectors.
>
> We hope the Reviewer finds these new Appendix sections to be an interesting and useful addition to the main text.
>
>
> As ever, please do not hesitate to contact us for any queries/clarifications.
>
>
> Best wishes,
>
>
> ICLR 2019 Conference Paper1058 Authors

---

### Official Review · AnonReviewer2 · 2018-11-02
**Interesting and simple idea**

**Rating:** 8
**Confidence:** 4

**Review:**

This submission presents a simple model for sentence representation based on max-pooling of word vectors. The model is motivated by fuzzy-set theory, providing both a funded pooling scheme and a similarity score between documents. The proposed approach is evaluated on sentence similarity tasks (STS) and achieves very strong performance, comparable to state-of-the-art, computationally demanding methods.

Pros:
+ The problem tackled by this paper is interesting and well motivated. Fast, efficient and non-parametric sentence similarity has tons of important applications (search, indexing, corpus mining).
+ The proposed solution is elegant and very simple to implement.
+ When compared to standard sentence representation models, the proposed approach has very good performance, while being very efficient. It only requires a matrix vector product and a dimension-wise max.
+ The paper is very well written and flows nicely.
+ Empirical results show significant differences between different word vectors. The simplicity of this approach makes it a good test bed for research on word vectors.

Cons:
- Nothing much, really.
- Eq. (3) is awkward, as it is a sequence of equalities, which has to be avoided. Moreover, if U is the identity, I don't think that the reader really need this Eq...

I have several questions and remarks that, if answered would make the quality of the presentation better:

* In infersent, the authors reported the performance of a randomly-initialized and max-pooled bi-lstm with fasttext vectors as the input lookup. This can be seen as an extreme case of the presented formalism, where the linear operator U is replaced by a complicated non linear function that is implemented by the random LSTM. Drawing that link, and maybe including this baseline in the results would be good.

* Related to this previous question, several choices for U are discussed in the paper. However, only two are compared in the experiments. It would be interesting to have an experimental comparison of:
- taking U = W
- taking U = I
- taking U as the principal directions of W
- taking U as a random matrix, and comparing performance for different output dimensions.

Overall, this paper is a very strong baseline paper. The presented model is elegant and efficient. I rate it as an 8 and await other reviews and the author's response.

---

> ### Author Response · Authors · 2018-11-12
> **Thank you!**
>
> Dear Reviewer,
>
> We would like to thank you for such a kind assessment of our work and so many positive comments.
> The Reviewer has asked some fascinating questions and we are jumping straight to them.
>
>
> * On InferSent
>
> Absolutely, in principle the linear operator U can be replaced by any non-linear function, such as a (deep) neural network. But because InferSent is a Bi-LSTM, the membership vector for a word w_t would depend on the membership vectors of words w_(t-1) and w_(t+1). By contrast, in our fuzzy *bag*-of-words model all the memberships vectors are computed separately and independently of each other.
> We genuinely feel these "fuzzy sequences" have a great research potential but have to leave them to future work.
>
> Randomly initialised InferSent uses GloVe vectors for its embeddings layer, followed by a randomly initialised Bi-LSTM. However, we see from [1] (Table 4) that its performance on STS14 is only 0.39, when averaged GloVe vectors already attain 0.54, while avg. fastText and word2vec both score above 0.63.
> In other words, random InferSent is very unlikely to be a good baseline for unsupervised semantic textual similarity. Of course, the trained InferSent is a very strong model and we already compare against it in Table 1.
>
> [1] Alexis Conneau, Douwe Kiela, Holger Schwenk, Loic Barrault, and Antoine Bordes.
> Supervised Learning of Universal Sentence Representations from Natural Language Inference Data. EMNLP 2017, pp. 670–680
>
>
> * Different choices for U (the universe matrix)
>
> We were very excited the Reviewer suggested the random matrix. We didn't mention this in the paper but in fact we tried all of the following universes:
>
> -------------------------------------------------------------------------------------
> GloVe|                                  STS12   STS13   STS14   STS15   STS16
> --------------------------------------------------------------------------------------
> Avg.                                        52.1      49.6      54.6       56.1      51.4
>
> W (top 100K)                         58.6     48.2       62.8       69.3      69.4
> DynaMax                               58.2     53.9       65.1       70.9      71.1
> Random 300x300	                57.0	     49.5       64.9       70.4     70.8
> Identity 300                           57.7     51.4       65.9       70.7      70.4
> SVD basis                               58.1     51.8       66.1       70.7      71.0
> SVD (top 200 vec)                 57.0     49.5       64.6       69.4      69.9
>
>
> For GloVe vectors the methods are basically the same but DynaMax gets good improvement over the max-pooled word vectors (identity matrix) with most other word vectors (Figures 1 & 2).
>
> We chose to focus on DynaMax and max-pooled vectors exclusively because only these 2 universes are non-parametric and deterministic.
> We ourselves feel that DynaMax is probably the strongest and safest choice overall for any kind of word vectors.
> However, it is sensible to start with just max-pooled word vectors because they avoid matrix multiplication altogether.
> We will be linking our code repository after the anonymity period and hope the community discovers universes that we haven't so far.
>
> Also, we agree that sequence of equalities in Eq. 3 is awkward, this is purely to emphasise the origins of max-pooled word vectors. We will consider how to alter this equation while keeping this message.
>
> Again, we would like to thank the Reviewer for such a positive assessment and great questions.
> Please do not hesitate to contact us for any further queries/clarifications.
>
>
> Best wishes,
>
>
> ICLR 2019 Conference Paper1058 Authors

---

> ### Author Response · Authors · 2018-11-27
> **Updates to the paper**
>
> Dear Reviewer,
>
> We wanted to let you know that the new version of the paper has now been uploaded.
> The main text remains almost the same, however, we have added 4 new sections to the Appendix.
>
> We included ablation studies on the DynaMax in Appendix C; in one of those we compare the dynamic universe, the identity matrix and also the random projection (as suggested by the Reviewer).
>
> Also, we added [1] to the citation list of papers that use the max-pooling operation.
>
>
> Finally, we added many other interesting results, including a novel significance analysis methodology for STS, a comparison between different fuzzy similarity coefficients and discussion on [0, 1]-fuzzy sets and their connection with normalised vectors.
>
> We hope the Reviewer finds these new Appendix sections to be an interesting and useful addition to the main text.
>
> [1] Alexis Conneau, Douwe Kiela, Holger Schwenk, Loic Barrault, and Antoine Bordes.
> Supervised Learning of Universal Sentence Representations from Natural Language Inference Data. EMNLP 2017, pp. 670–680
>
>
> As ever, please do not hesitate to contact us for any queries/clarifications.
>
>
> Best wishes,
>
>
> ICLR 2019 Conference Paper1058 Authors

---

### Official Review · AnonReviewer3 · 2018-11-06
**Very polished paper, simple but effective.**

**Rating:** 8
**Confidence:** 3

**Review:**

This is one of the best papers I reviewed so far this year (ICLR, NIPS, ICML, AISTATS), in terms of both the writing and technical novelty.

Writing: the author provided sufficient context and did comprehensive literature survey, which made the paper easily accessible to a larger audience. And the flow of this paper was very smooth and I personally enjoyed reading it.

Novelty: I wouldn't say this paper proposed a groundbreaking innovation, however, compared to many other submissions that are more obscure rather than inspiring to the readers, this paper presented a very natural extension to something practitioners were already very familiar with: taking an average of word vectors for a sentence and measure by cosine similarity.  Both max pooling and Jaccard distance are not something new, but the author did a great job presenting the idea and proved it's effectiveness through extensive experiments. (disclaimer: I didn't follow the sentence embedding literature recently, and I would count on other reviewers to fact check the claimed novelty of this paper by the authors)

Simplicity: besides the novelty mentioned above, what I enjoyed more about this paper is it's simplicity. Not just because it's easy to understand, but also it's easy to be reproduced by practitioners.

Quibbles: the authors didn't provide error bar / confidence interval to the results presented in experiment session. I'd like to know whether the difference between baselines and proposed methods were significant or not.

Miscellaneous: I have to say the authors provided a very eye-catching name to this paper as well, and the content of the paper didn't disappoint me neither. Well done :)

---

> ### Author Response · Authors · 2018-11-13
> **Thank you!**
>
> Dear Reviewer,
>
> We would like to thank you for such a positive assessment of our work.
> We were especially thrilled the Reviewer found our paper to be among the best they reviewed this year.
>
> Regarding the significance analysis, unfortunately, the SentEval toolkit [1] we're using does not support this functionality.
> Moreover most works known to us, including some of the most prominent works published at ICLR, do not conduct significance analysis for STS benchmarks ([2], [3], [4], [5]). Admittedly, some other works apply the Fisher's z test, which we believe is not appropriate in this setting. More appropriately, some apply the William's t test [6] or Steiger's z test [7] for correlated correlations. To the best of our knowledge, these tests require that data comes from a normal distribution, which is not case for STS. Although we have done a similar analysis using Steiger's z, we have to refrain from reporting (potentially) statistically invalid results and are looking to obtain further evidence that these tests can in fact be applied here. We are also looking into alternative (non-parametric) methods and will let the Reviewer know when our analysis is complete.
>
> [1] Alexis Conneau and Douwe Kiela (2018). SentEval: An Evaluation Toolkit for Universal Sentence Representations. http://arxiv.org/abs/1803.05449
> [2] John Wieting, Mohit Bansal, Kevin Gimpel and Karen Livescu. ICLR 2016.
> [3] Sanjeev Arora, Yingyu Liang and Tengyu Ma. A Simple but Tough-to-Beat Baseline for Sentence Embeddings. ICLR 2017.
> [4] Jiaqi Mu and Pramod Viswanath. All-but-the-Top: Simple and Effective Postprocessing for Word Representations. ICLR 2018.
> [5] Sandeep Subramanian Adam Trischler, Yoshua Bengio and Christopher J Pal. Learning General Purpose Distributed Sentence Representations via Large Scale Multi-task Learning. ICLR 2018.
> [6] Williams, E. J. (1959). The comparison of regression variables. Journal of the Royal Statistical Society, Series B, 21, 396-399.
> [7] Steiger, J. H. (1980). Tests for comparing elements of a correlation matrix. Psychological Bulletin, 87(2), 245-251.
>
>
> Again, thank you very much and please do not hesitate to contact for with any further queries/clarifications.
>
>
> Best wishes,
>
>
> ICLR 2019 Conference Paper1058 Authors

---

> ### Author Response · Authors · 2018-11-27
> **Updates to the paper**
>
> Dear Reviewer,
>
> We wanted to let you know that the new version of the paper has now been uploaded.
> The main text remains almost the same, however, we have added 4 new sections to the Appendix.
>
> As promised, we have now conducted the significance analysis of the results and included our findings in the paper (Intro + Appendix D).
> In summary, we found that recent literature on STS tends to apply unspecified or inappropriate parametric tests, or leave out significance analysis altogether in the majority of cases.
> We propose to construct nonparametric bootstrap confidence intervals with bias correction and acceleration. These intervals have much milder assumption on the test statistic than the parametric tests.
> To the best of our knowledge, such methodology has not been applied to the STS benchmarks before and can be viewed as an additional yet important contribution of our work. We hope the Reviewer and the community find our analysis useful and interesting. It was also good fun for us! Thanks for bringing this up.
>
> We also added many other interesting results including ablation studies on DynaMax, a comparison between different fuzzy similarity coefficients and discussion on [0, 1]-fuzzy sets and their connection with normalised vectors.
> We hope the Reviewer finds these new Appendix sections to be an interesting and useful addition to the main text.
>
>
> As ever, please do not hesitate to contact us for any queries/clarifications.
>
>
> Best wishes,
>
>
> ICLR 2019 Conference Paper1058 Authors

---

### Public Comment · (anonymous) · 2018-10-22
**About the matrix U**

This work is very good, I am very interested. Fuzzy set theory is a very effective tool, which can explain and describe the uncertainty of data. The matrix U in this paper is very important, indicating the degree of membership of the elements in the fuzzy set, so the element value in U should be [0,1]. However, when U=W (W represents the word representation), the elements are a real number, which can be positive or negative. If U is unconstrained, the overall interpretability will be discounted.
In addition, in the experimental part, if the author can give an example of U, it will be more perfect.

---

> ### Author Response · Authors · 2018-10-23
> **Good question, thanks!**
>
> Thank you for your kind feedback and interest in our paper.
> We also believe fuzzy set theory is a very useful framework and we were excited it worked well in this setting.
>
> Onto your question, we briefly discuss the choice of R as opposed to [0, 1] in Sections 2.1 and 2.2.4 but will extend these sections in a revised version of the manuscript.
>
>
> The universe matrix U is a (K x 300) matrix, i.e. the universe contains K entities and each row of U is an embedding of a single entity. When U=W, the entities are words and the rows of U are simply the word embeddings.
>
> Now we want to convert a singleton {w} into a fuzzy set. We compute the membership values:
>
> mu = [sim(w, u1), sim(w, u2), ..., sim(w, uK)]
>
> We see that the membership values actually come from the similarity function sim(w, u) and not from the matrix U directly, so what's really important is whether values of sim(w, u) are in [0, 1] or R.
>
> In our work, sim(w, u) is the dot product w * u, which does indeed take any real value. Below we discuss why dot product is a reasonable choice.
>
> 1.
> Intuitively, it's all the same up to the scale. We can easily map any real number into (0, 1) using, e.g. the
> logistic function s(x) = 1/(1+e^-x) and vice versa. So it's really not that important whether the values are in (0, 1)
> or in R. In fact, there are many various extensions of fuzzy set theory that work with more general ranges than [0,1].
> For a quick reference: https://en.wikipedia.org/wiki/Fuzzy_set#Extensions
>
> 2.
> The membership function for multisets (bags) takes values in N (i.e. the non-negative integers), so these values are already outside [0, 1]. We see that the standard [0, 1]-fuzzy sets are incompatible with multisets, that's why we constructed the most general sensible thing. Sets, bags, and [0,1]-fuzzy sets are all just a special case of fuzzy BoW.
> Interestingly, since we always max-pool with a zero vector, fuzzy BoW will not contain any negative membership values.
> This was not our intention, just a by-product of the model. As discussed in 1., negative values are fine. They just mean
> the element is "really" not in the set.
>
>
> 3.
> Still, why did we choose dot product and not, say, sim(w, u) = max(cos(w, u), 0)?
> We know that cosine similarity is in [-1, 1], so sim(w, u) will be in [0, 1].
> One reason is that word vectors are usually trained to maximise some kind of dot product in their objectives.
> A more practical reason is simply because dot product works much better.
>
> Turns out, if we normalise word embeddings we will get the same fuzzy BoW as if we used max(cos(w, u), 0)
> (simply because dot product is the same as cosine similarity for normalised vectors).
>
> Below we give the results for GloVe vectors
>
> 			         STS12	  STS13	  STS14	  STS15	  STS16
>
> Avg.		         52.1	   49.6	   54.6	   56.1	   51.4
> Avg. norm	         47.1	   44.9	   49.7	   52.0	   44.0
>
> DynaMax	         58.2	   53.9	   65.1	   70.9	   71.1
> DynaMax norm	 53.7	   47.8	   59.5	   66.3	   62.9
>
> We see here that normalisation hurts both averaged word vectors and well as max-pooled word vectors.
> However, DynaMax norm still significantly outperforms Avg. norm. It even outperforms Avg. without norm on all but one tasks.
>
> Hopefully, the above taken together explains why the membership values can indeed be any real numbers.
> Again, thank you very much for a good question.
> We will add this discussion as well as results for more word vectors into a separate section in the Appendix.
>
> Please do not hesitate to ask us any additional questions.
>
> Best wishes,
>
> ICLR 2019 Conference Paper1058 Authors

---

> ### Author Response · Authors · 2018-11-27
> **Updates to the paper**
>
> Dear Reader,
>
> We wanted to let you know that the new version of the paper has now been uploaded.
>
> As promised, we have included a detailed discussion on [0,1] vs R in Appendix A as well as results for more word vector types. The discussion is a distilled version of what we already wrote here; nevertheless we hope the Reader finds it useful to see this discussion in the context of the entire paper.
> Again, thanks a lot for your suggestion.
>
> As ever, please do not hesitate to contact us for any queries/clarifications.
>
>
> Best wishes,
>
>
> ICLR 2019 Conference Paper1058 Authors

---

### Author Response · Authors · 2018-10-24
**Small typo in Algorithm 1**

Dear Readers,

We have spotted a small typo in the manuscript.
In Algorithm 1, Line 3 the vector with all zeros z should have dimensions 1 x (k+l) and not 1 x d.
The dimension of the zero vector has to match the dimension of other vectors in the max-pooling operation, all of which are 1 x (k+l) after the projection onto U.
Alternatively, we can keep zero vector to be 1 x d but then we need to project it as well in Lines 8 and 9, i.e. compute zU^T.
However, the latter would be a useless computation, so we prefer the first option.

We apologise for any inconveniences this typo might have caused and will fix it in the next version of the manuscript.


Best wishes,


ICLR 2019 Conference Paper1058 Authors

---

> ### Author Response · Authors · 2018-11-27
> **Fixed**
>
> This typo has been fixed in the updated version of the paper.

---

### Author Response · Authors · 2018-11-09
**Please stay tuned**

Dear Reviewers and Readers,

We were absolutely thrilled that our work received such a positive assessment.
We would like to apologise for the delay in our replies; we are in fact working very hard to run additional analyses to quantitatively support our replies to each Reviewer.

These include :
- significance tests (Reviewer 3)

- comparisons of different choices for the universe matrix U and a short story on how we arrived at each of them (Reviewer 2)

- certain experiments to support our response to Review 1

We expect to post very detailed replies by Monday at the latest. We hope you all have a nice weekend and please stay tuned.


Best wishes,


ICLR 2019 Conference Paper1058 Authors

---

### Author Response · Authors · 2018-11-27
**Updates to the paper**

Dear Reviewers and Readers,

We wanted to let you know that the new version of the paper has now been uploaded.
Here is a short summary of the changes.

1. The main text remains almost the same. We have fixed some typos, added citations and made some statements a bit clearer.

2. We have added a novel significance analysis of our results in Appendix D.
We found that the majority of recent literature on STS either uses inappropriate or unspecified parametric tests or leaves out significance analysis altogether. We propose to construct nonparametric bootstrap confidence intervals with bias correction and acceleration. These intervals have much milder assumption on the test statistic than the parametric tests. To the best of our knowledge, such methodology has not been applied to STS benchmarks before and can be viewed as an additional contribution of our work to the community.

3. We conduct ablation studies on our best-performing algorithm, DynaMax Jaccard, in Appendix C. In particular, we play with different universes, pooling schemes, and similarity functions. We find that DynaMax Jaccard remains the winner and conclude that the three components - the dynamic universe, the max-pooling operation and the fuzzy Jaccard index - all contribute to the strong performance of the model.


4. We compare fuzzy set similarity measures derived from Jaccard, Otsuka-Ochiai and Sørensen–Dice coefficients and show they have almost identical performance across tasks and word vectors, quantitatively confirming that our results are in no way specific to the Jaccard index.


5. We discuss the difference between [0,1] and real-valued membership functions in Appendix A. In particular, we show that one simple way to construct [0,1]-fuzzy sets is to simply normalise the vectors. However, normalisation hurts both our method and the baseline (because it renders all the words equally important) and should generally be avoided in this setting.



We hope to have addressed the Reviewers' questions and concerns. We also hope the Reviewers and Readers will find the new Appendix sections to be a useful and interesting addition to the main text.


Best wishes,


ICLR 2019 Conference Paper1058 Authors

---

### Public Comment · (anonymous) · 2018-12-10
**Add more comparisons to Table 2?**

Very interesting paper!

I was wondering why you left out the results from uSIF (Ethayarajh, 2018) in your Table 2, despite briefly citing it earlier on. avg-uSIF+PCA -- which the original paper denotes as UP -- looks like it gets much better results on the STS tasks than DynaMax-SIF (see Table 1 in (Ethayarajh, 2018)). For example, DynaMax-SIF gets 61.1 with GloVe for STS'12 while uSIF+UP gets 64.9 with GloVe for STS'12.

It looks like your method improves anything it's applied to, so I would suggest doing DynaMax-uSIF and then reporting those results in Table 2 as well -- I imagine they would be even better than what you have now.

---

> ### Author Response · Authors · 2018-12-13
> **Good question! Answer: no need because uSIF is NOT better than SIF.**
>
> Dear Reader,
>
> Thank you very much for your interest in our paper and a very good question.
>
> Ethayarajh (2018) discovered a very clever method to estimate the weight parameter "a" in an unsupervised way, improving upon Arora et al. (2017), who technically had to estimate the same parameter on the training set. We cited Ethayarajh (2018) precisely for this neat and important contribution to the community.
>
> However, we must inform the Reader that uSIF model is NOT any better than SIF of Arora et al. (2017).
> The numbers in Ethayarajh (2018) are correct, but all the improvements are due to post-processing tricks and differences in experimental setups.
>
> We conducted an ablation study using the codebase released by Ethayarajh (2018): https://github.com/kawine/usif
> and GloVe vectors http://nlp.stanford.edu/data/glove.840B.300d.zip
>
> SIF weights:            a/(a + p_w),              a=1.0e-3
> uSIF weights:          a/(a/2 + p_w),          a derived automatically, approx. 1.2e-3
>
>
>                     STS12    STS13    STS14    STS15
>
> SIF                60.1      60.2       66.5       62.9
> uSIF              60.4      60.6       67.0       63.6
>
> Table 1: SIF vs uSIF; no PC removal, no custom norm
>
>
>                                         STS12    STS13    STS14    STS15
>
> SIF  +1PC +norm           64.6       70.9       73.7       75.2
> SIF  +5PC +norm           64.9       71.8       74.4       76.3
> uSIF +5PC +norm          64.9       71.7       74.4       76.1
>
> Table 2: SIF vs uSIF, +PC removal, +custom norm
>
>
> On both occasions, uSIF was not better than SIF.
> Moreover, removing 5 PCs as in Ethayarajh (2018) leads to only marginal (if significant) improvement over removing just 1 PC as in Arora et al. (2017).
> However, the actual sources of improvement are worth discussing
>
> - Text pre-processing and custom codebase
>
> We rely on the established SentEval toolkit (Conneau & Kiela, 2018) for all the evaluations in our paper (including prior work). A lot of improvement reported by Ethayarajh (2018) is due to a custom codebase, therefore it's inappropriate to say uSIF+PCA is "better".
>
>
> - PC removal
>
> Neither it is appropriate to compare DynaMax with methods that run PCA on the whole test set as in Arora et al. (2017) and Ethayarajh (2018). They are simply different in nature. DynaMax is a similarity measure between 2 sentences, it doesn't require anything beyond that. PCA-based embeddings are "fitted" to a concrete dataset. They can be used for things like clustering but not for on-the-fly processing of incoming queries.
>
>
> - Custom normalisation scheme
>
> The normalisation scheme proposed by Ethayarajh (2018) is not derived from the modelling assumptions of uSIF, therefore it's a post-processing trick (just like PCA). We see no issues with this scheme per se, but we can hardly justify its use in the present work.
>
>
> Again, we would like to thank the Reader for bringing up a good and important question.
> There were a lot of comments to other OpenReview submissions this year asking to compare against uSIF.
> Based on our careful analysis, we believe including DynaMax + uSIF in our tables adds no value.
> We hope our reply is acceptable to the Reader and useful to the community.
>
> Please do not hesitate to contact us for any further queries/clarifications.
>
>
> Sanjeev Arora, Yingyu Liang and Tengyu Ma. A Simple but Tough-to-Beat Baseline for Sentence Embeddings. ICLR 2017.
> Alexis Conneau and Douwe Kiela. Senteval: An evaluation toolkit for universal sentence representations. arXiv preprint arXiv:1803.05449, 2018.
> Kawin Ethayarajh. Unsupervised Random Walk Sentence Embeddings: A Strong but Simple Baseline. Rep4NLP@ACL 2018: 91-100
>
>
> Best wishes,
>
>
>
> ICLR 2019 Conference Paper1058 Authors

---

### Author Response · Authors · 2019-01-11
**Camera ready version**

Dear Area Chair, Reviewers and Readers,

We were delighted to learn that our work has been recommended for acceptance and are looking forward to presenting it at the conference. We have now uploaded the camera ready version of the manuscript and linked the source code repository.

We were absolutely thrilled by the amount of positive feedback our work has received and would like to thank everyone who participated in this forum.


Best wishes,


ICLR 2019 Conference Paper1058 Authors

---

### Meta-Review · Area_Chair1 · 2018-12-14
**The simplicity is what makes the proposed methods elegant. The empirical results are strong.**

**Confidence:** 4
**Recommendation:** Accept (Poster)

**Metareview:**

This paper presents new generalized methods for representing sentences and measuring their similarities based on word vectors. More specifically, the paper presents Fuzzy Bag-of-Words (FBoW), a generalized approach to composing sentence embeddings by combining word embeddings with different degrees of membership, which generalize more commonly used average or max-pooled vector representations. In addition, the paper presents DynaMax, an unsupervised and non-parametric similarity measure that can dynamically extract and max-pool features from a sentence pair.

Pros:
The proposed methods are natural generalization of exiting average and max-pooled vectors. The proposed methods are elegant, simple, easy to implement, and demonstrate strong performance on STS tasks.

Cons:
The paper is solid, no significant con other than that the proposed methods are not groundbreaking innovations per say.

Verdict:
The simplicity is what makes the proposed methods elegant. The empirical results are strong. The paper is worthy of acceptance.